# Protein engineering using variational free energy approximation

Evgenii Lobzaev [1,2], Michael A. Herrera[1], Martyna Kasprzyk [1] &
Giovanni Stracquadanio [1] ✉

Engineering proteins is a challenging task requiring the exploration of a vast design space. Traditionally, this is achieved using Directed Evolution (DE), which is a laborious process. Generative deep learning, instead, can learn biological features of functional proteins from sequence and structural data-sets and return novel variants. However, most models do not generate thermodynamically stable proteins, thus leading to many non-functional variants. Here we propose a model called PRotein Engineering by Variational frEe eNergy approximaTion (PREVENT), which generates stable and functional variants by learning the sequence and thermodynamic landscape of a protein. We evaluate PREVENT by designing 40 variants of the conditionally essential *E. coli* phosphotransferase N-acetyl-L-glutamate kinase (*Ec*NAGK). We find 85% of the variants to be functional, with 55% of them showing similar growth rate compared to the wildtype enzyme, despite harbouring up to 9 mutations. Our results support a new approach that can significantly accelerate protein engineering.

Designing a new protein entails the search for novel, stable and functional amino acid sequences that integrate seamlessly with an existing protein fold, while minimizing the associated free energy of the new configuration[1]. In practice, this requires evaluating thousands of sequence variants[2], derived from a suitable protein template, for both their thermodynamic and operational fitness, often with respect to a desired downstream application, e.g., biocatalysis or enzyme replacement therapy. This is conventionally achieved experimentally using Directed Evolution (DE)[3], where thousands of protein variants are synthesised, screened and selected through a selection process that can converge to a sequence optimum that meets specific operational requirements. However, DE is a laborious and expensive process, which requires a high degree of laboratory automation to achieve the throughput required by the industrial biotechnology sector.

Computational methods instead hold the promise to speed up protein engineering by providing a way to screen in silico favourable mutations by biophysically modelling thermodynamic changes by amino acid changes[4,5]. While a plethora of biophysical methods have been developed to facilitate the process of protein engineering, they are often constrained by the computational cost of free energy calculations, which in turn limits the exploration of the protein design space. Recently, generative deep learning has emerged as a promising tool for protein engineering, as it can learn the biological properties associated with functional proteins using large sequence and structural datasets[6–10]. Nonetheless, while the generated sequences broadly resemble those observed in Nature, it remains challenging to obtain functional designs, since most methods cannot condition the sequence generation process towards thermodynamically stable variants.

Here we hypothesised that we could overcome current limitations of generative protein engineering models by learning the sequence-to-free-energy relationship from a small set of biophysical simulations over a library of computationally designed variants using a Variational Autoencoder (VAE)[11]. To this end, we developed the PRotein Engineering by Variational frEe eNergy approximaTion (PREVENT) model, which allows both the controlled generation of variants with minimal free energy as well as the prediction of the free energy associated with generated variants for downstream experimental prioritization.

[1]School of Biological Sciences, The University of Edinburgh, Edinburgh, United Kingdom. [2]chool of Informatics, The University of Edinburgh, Edinburgh, United Kingdom. ✉e-mail: giovanni.stracquadanio@ed.ac.uk

We evaluated PREVENT by designing and experimentally testing variants of the archetypal *E. coli* phosphotransferase N-acetyl-L-glutamate kinase (*Ec*NAGK)[12]. *Ec*NAGK is a small, 258 amino acid long conformationally dynamic homodimer, comprising a bi-domain architecture organised in a Rossmann-like ($\alpha/\beta/\alpha$) sandwich. In Nature, it is primarily responsible for ATP-dependent phosphorylation of N-acetyl-L-glutamate (NAG) in the L-arginine biosynthesis pathway. The intrinsic flexibility of *Ec*NAGK, as evidenced by several crystal structures (PDB: 1GS5, 2WXB, 1OHA, 1OHB, 1OH9), is hypothesised to be crucial for catalysis and hints towards a complex thermodynamical landscape. Using our model, we designed a library of 40 new *Ec*NAGK variants and observed that 85% of the transformed variants could substitute for the wildtype enzyme despite harbouring up to 9 mutations compared to the wildtype.

Taken together, our results support a new approach to generative protein design that can dramatically accelerate engineering of novel, functional proteins.

## Results

L-arginine is a ubiquitous proteinogenic amino acid necessary for the basic physiological function of all organisms. De novo L-arginine biosynthesis proceeds via a critical L-ornithine precursor in both animals and bacteria. Distinctly from animals, however, bacterial L-ornithine is generated from a cascade of N-acetylated, rather than non-acetylated, intermediate (see Supplementary Fig. 1)[13].

The first committed step in *E. coli* L-ornithine biosynthesis is the transacetylation of L-glutamate using acetyl-CoA to yield NAG, catalysed by N-acetyl-L-glutamate synthase (NAGS). In the second biosynthetic step, the nascent NAG is phosphorylated to furnish N-acetyl-L-glutamyl 5-phosphate (NAGP); this ATP-dependent conversion is catalysed by aforementioned *Ec*NAGK, encoded in *E. coli* by the *argB* gene. Importantly, the binding of ATP to *Ec*NAGK triggers a major inter-domain conformational shift that greatly tightens the active site, thereby permitting the facile phosphoryl transfer from ATP to NAG. This reaction is further facilitated by invariant residues Lys8, Gly11, Gly44, Gly45 and Lys217, which work to correctly orient the substrates and stabilise the transition state. Following phosphoryl transfer, the enzyme relaxes into an open conformation and liberates the products NAGP and ADP in preparation for the next catalytic cycle.

Here we used our PREVENT model (see Supplementary Fig. 2) to replace the wildtype *Ec*NAGK enzyme with new, unseen variants thereof. To do that, we began by performing in silico mutagenesis of the *Ec*NAGK wildtype sequence (Uniprot id: P0A6C8; PDB id: 1GS5) to build the input dataset required to approximate the sequence-to-energy relationship for this enzyme (see "Methods"). With our procedure, we generated 117,387 unique variants, with each variant harbouring up to 38 mutations ($\approx 15\%$ of the residues), and then computed the associated free energy, $\Delta G$, using FOLDX[14]. Each position of the wildtype enzyme, except the first one, is mutated on average in 6.58% of the variants; as expected, most of the introduced mutations are destabilizing, with a significant positive correlation between number of mutations and free energy.

We then split the dataset into a training and validation sets (90%, 106,238 variants) (see Fig. 1A, B) and a held-out test set (10%, 11,149 variants) (see Supplementary Fig. 3) to evaluate the performance of our model. We maintained a similar $\Delta G$ distribution in the training and test sets for a fair comparison.

In order to study how PREVENT performances depend on the size of the input mutagenesis dataset, we further subsampled the initial training dataset by selecting 25K, 50K and 75K sequences at random, albeit maintaining a $\Delta G$ distribution similar to the original dataset.

Using these datasets, we then trained our default transformer VAE model (see "Methods") with 128-dimensional latent space for 1400 epochs using mini batches of 256 sequences, and by setting the learning rate to $10^{-4}$ and the dropout probability to 0.2 for regularization. Moreover, to learn effective latent space encodings, we masked 25% of each sequence residues at random. With this experimental setup, we tested our model on 4 different configurations.

We first assessed model performances using a held-out test set by looking both at the sequence reconstruction error and the Root Mean Square Error (RMSE) of the free energy across the 4 experiments. After attaining the prefixed number of epochs, the reconstruction error was 114.80 (perplexity: 0.66) for the full dataset, suggesting that our model is able to effectively reconstruct sequences from the latent space, a trend that was consistent regardless of the size of the training dataset (see Supplementary Table 1). We then looked at the accuracy of the predicted free energy, and obtained the best results when using the full dataset (RMSE = 9.27); nonetheless, even when using only 25K sequences, the model achieves comparable performances (RMSE = 12.12). Importantly, while the predicted energy values might differ from those estimated by FOLDX, they are strongly correlated, with Spearman correlation ranging from $\rho = 0.96$ when training on the full dataset and only dropping to $\rho = 0.92$ when using only 25K sequences, suggesting that predicted values are a robust metric for variants ranking. This is further confirmed by the concordance at the top (CAT) analysis of variants ranked by free energy values; in particular, we found a concordance higher than 80% when looking at the top 1000 ranked sequences, a trend observed regardless of the size of the mutagenesis dataset used (see Fig. 1C, D).

We then analyzed the *Ec*NAGK encodings to check whether PREVENT would map different variants to different regions of the latent space. To achieve this, we computed the empirical variance of each latent component, defined as $\text{diag}(\text{cov}_x \mathbb{E}_{q_\phi(z|x)}[z])$[15], and considering a component as being active if its standard deviation was greater than 0.1. Here we found that the vast majority (126) of the 128 components to be active, albeit to a different level, suggesting that our model maps variants to different regions of the latent space. We then proceeded with the analysis of the latent space organization by performing Principal Component Analysis (PCA) of the expected value of the variational posterior distribution $q_\phi(z|x)$ for the sequences in the mutagenesis dataset, which revealed a structural organization, where sequences with lower free energy were located in the central part of the latent space, whereas sequences with higher $\Delta G$ values were located in the periphery, remotely resembling a folding funnel (see Fig. 2).

Taken together, we found that PREVENT is able to effectively reconstruct *Ec*NAGK sequences and the expected free energy from the latent space.

### Hyperparameter analysis

We also performed a hyperparameter analysis to understand how the choice of the latent dimension, the transformer size and the number of training epochs affect the performance of PREVENT on the held-out test set. We used our original transformer model configuration as well as another, twice as small, configuration with 4 encoder layers, 2 decoder layers, 256 embedding size and 4 heads. First, we noticed that training for 500 epochs is usually sufficient to obtain a minimal validation loss, as most models converge between 100 and 200 epochs. Then, we confirmed that the choice of the latent dimension does not affect sequence reconstruction error, regardless of the size of the model and latent space. However, the model size and the latent dimension do affect the RMSE of the free energy, with the smaller model with 128-dimensional latent space achieving the lowest RMSE of 6.84 across all experiments. Nevertheless, the Spearman correlation remained high across all models, ranging from 0.97 to 0.98, which is comparable to the original model results, used in all downstream tasks (see Supplementary Table 2).

 

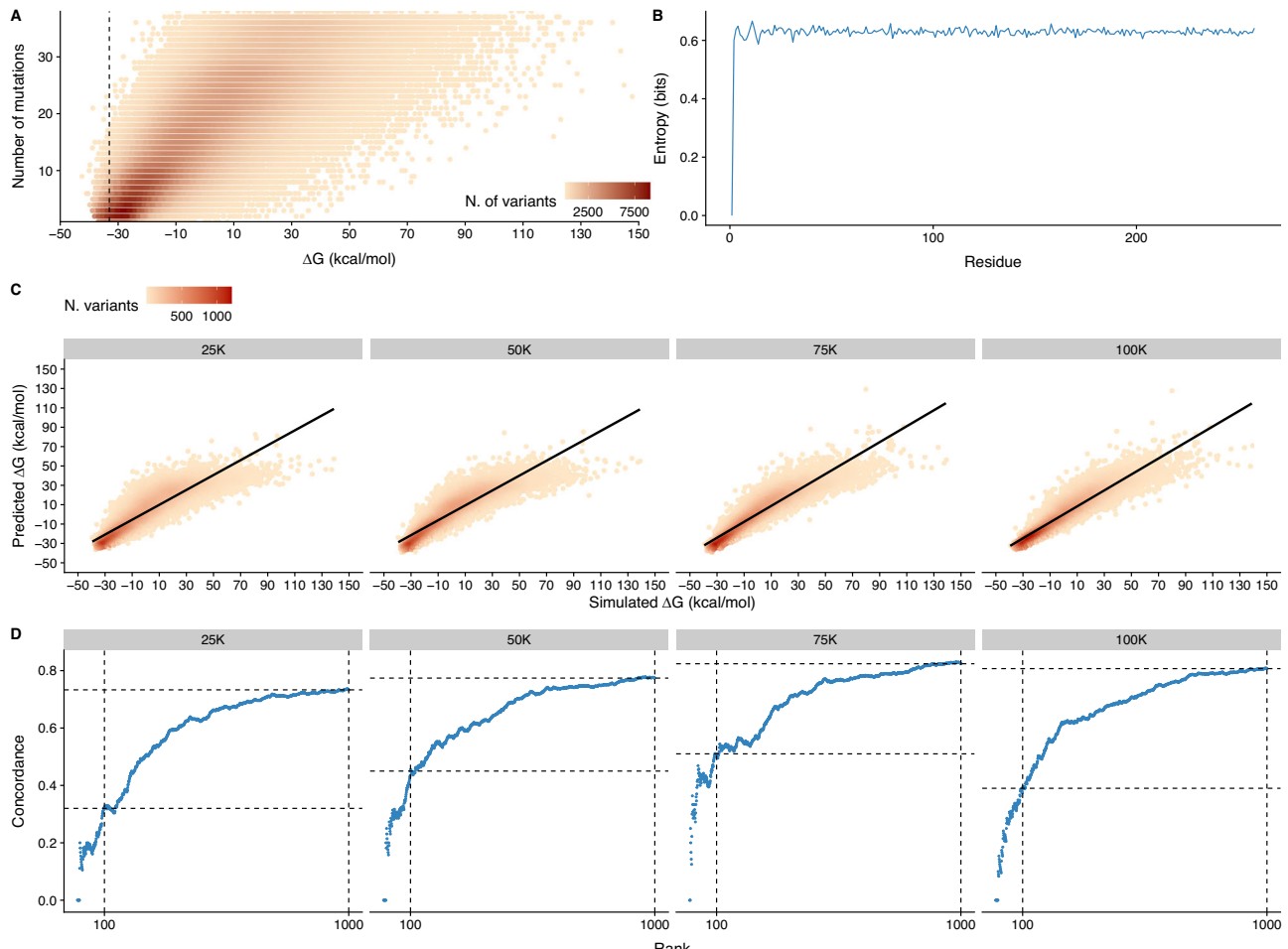

**Fig. 1 | PREVENT performance analysis. A** Thermodynamic landscape approximation ($\Delta G$) as a function of the number of mutations in $Ec$NAGK variants in the training dataset. Black dashed line denotes the free energy of the wildtype $Ec$NAGK. **B** Amino acid entropy of $Ec$NAGK variants in the training dataset. Every position, except the first methionine, is mutated in 6.58% of the generated variants. **C** PREVENT estimates of $\Delta G$ values against free energy estimates simulated using FOLDX for different training set sizes. RMSE is 9.28 for 100K training set, 9.74 for 75K training set, 11.24 for 50K training set and 12.12 for 25K training set. Spearman correlation coefficient is 95.53% for 100K training set, 95.25% for 75K training set, 93.58% for 50K training set and 91.79% for 25K training set. **D** concordance at the top (CAT) analysis of sequences ranked using free energy information. PREVENT achieves a concordance 75% with FOLDX simulations regardless of the size of the training set, albeit moderate differences in ranking top sequences are observed when considering the top 100 and 1000 variants.

## Sequence analysis of the $Ec$NAGK variants generated by PREVENT

We then proceeded to perform sequence analysis of variants generated by the two PREVENT strategies, namely prior optimization of free energy (POE) and seeded non optimal free energy ranking (SNE). We generated 2000 variants using POE and 30,000 samples for SNE, and then removed those sequences that were i) present in the training set, ii) have mutations in the binding site, or iii) are duplicated.

With this criteria, we obtained 1282 new variants (66%) for POE and 13,323 variants for SNE (44%), which is expected given that the first strategy freely explores the free energy landscape of $Ec$NAGK, whereas the second generates sequences within the region of the wildtype sequence. Importantly, POE variants are more diverse (see Fig. 3A) than SNE, with an average of 7.09 and 3.57 mutations respectively. Surprisingly, we observed that only ≈ 25% of mutations were conservative, as measured by BLOSUM62 substitution matrix; this result suggests that PREVENT might select energetically favorable mutations regardless of evolutionary information (see Fig. 3B). Finally, while the type of mutations was highly similar across strategies, they are located differently across the wildtype $Ec$NAGK sequence (see Fig. 3C); in particular, POE predominantly mutates the C-terminus harboring the ATP-binding domain of the protein, which harbours highly non

conservative mutations, such as the tryptophan introduced at position 212 in place of the aspartic acid residue (see Fig. 3D), whereas SNE introduced mutations almost uniformly across the wildtype sequence (see Fig. 3E).

Taken together, we found that PREVENT can generate highly diverse protein variants, which carry not only conservative mutations but also non conservative ones, which would not be selected by using homology information alone.

## Design of a library of $Ec$NAGK variants

We then built a library of $Ec$NAGK variants for experimental validation, in order to estimate the expected fraction of functional variants generated by PREVENT.

To do that, we took the 10 variants with lowest free energy obtained with the POE strategy and the 10 variants with the lowest free energy generated with the SNE strategy. We further augmented this library by adding 10 variants with the highest ELBO, denoted as SNL, and 10 with the highest identity with respect to the wildtype, hereby denoted as SNI, in order to compare our energy-driven design against common generative strategies for protein engineering (see Fig. 4A). Variants in the library have an average free energy ranging from -29.8 kcal/mol for the SNE group, to -18 kcal/mol for the SNL group, with a

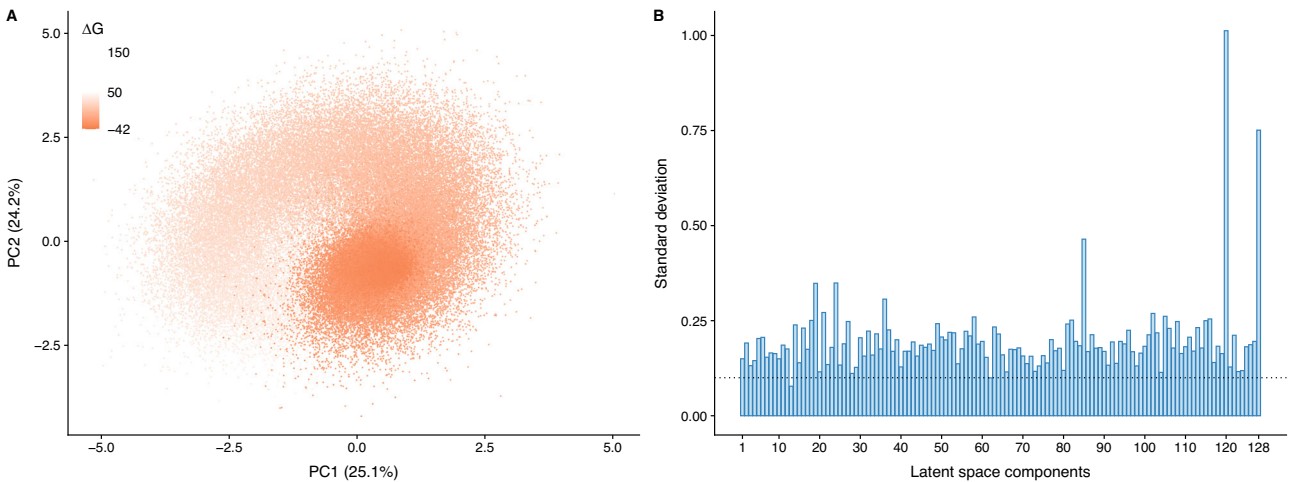

**Fig. 2 | PREVENT latent space analysis. A** Principal component analysis of the $\mathbb{E}_{q_\phi(z|x)}[z]$ embeddings generated by PREVENT for the sequences in the training set and color coded according to the corresponding free energy $\Delta G$ values. **B** Activation plot for the 128-dimensional latent space learned. The x-axis represents each latent component and y-axis the standard deviation of the corresponding value learned during training; the dashed line represents the expected value of each component under a null model of random value assignment.

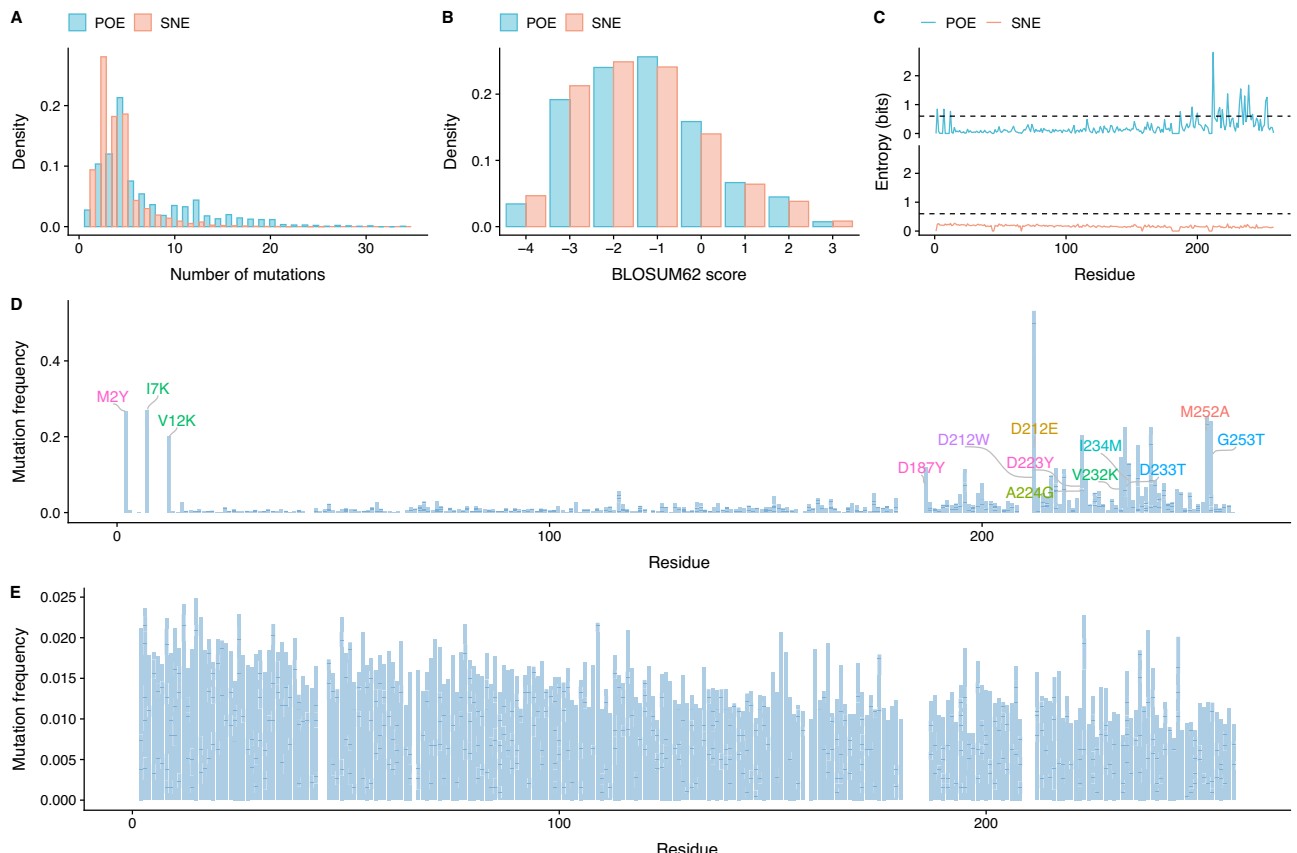

**Fig. 3 | Sequence analysis of PREVENT generated variants. A** Distribution of the number of mutations in variants generated using the POE and SNE strategies; the POE strategy generates more diverse variants compared to SNE. **B** Distribution of the BLOSUM62 substitution matrix scores for the 9083 mutations introduced by the POE strategy and for the 47,573 mutations introduced by the SNE strategy; the vast majority of the mutations are non-conservative. **C** Residue level entropy for variants obtained using POE and SNE strategies; POE preferentially adds mutations at the C-terminus of the protein. Most frequent mutations in variants generated by the POE strategy (**D**) and the SNE strategy (**E**). While SNE variants have a similar mutational profile to the training set, POE variants have a vastly different mutational profile, with surprisingly non-conservative mutations highly represented in all variants.

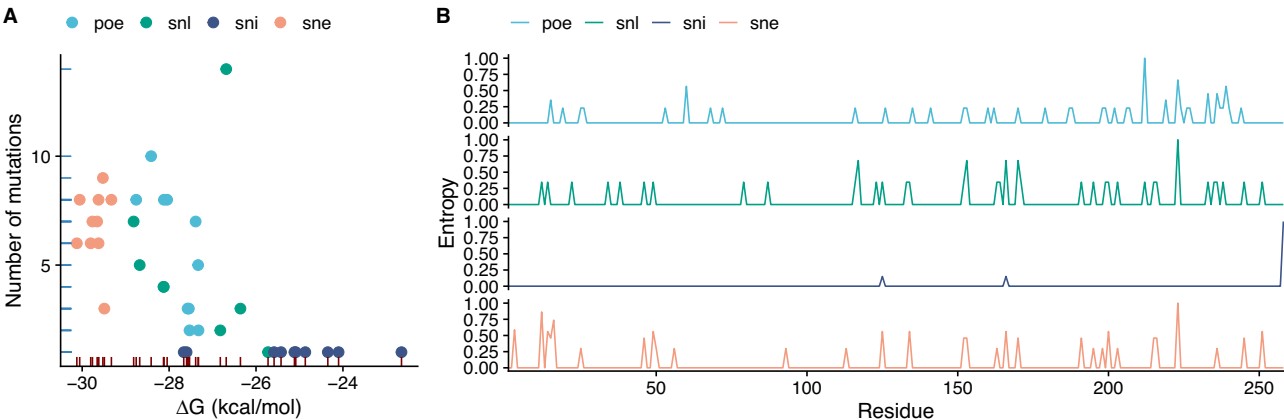

**Fig. 4 | Design of a library of 40 *Ec*NAGK variants using PREVENT. A** Free energy and number of mutations of each variant colored by design strategy. **B** Normalized sequence entropy of variants generated using different design strategies; energy-based strategies (POE, SNE) introduce mutation preferably in the C-terminus of the wildtype protein.

number of mutations ranging from 6.8 for the SNE group to 1 for the SNL group, mostly located in the C-terminus of the protein for the variants generated by energy-driven strategies (see Fig. 4B). Taken together, we screened a library of 40 new and diverse variants with a significant variability in free energy.

Before proceeding with experimental work, we performed a series of quality control steps, first, by comparing the predicted free energy values of each variant to the FOLDX estimates where, as expected, we found a strong correlation between the two estimates (Spearman correlation; $\rho = 0.77$, *p*-value: $5.7 \times 10^{-9}$). We additionally checked whether the correlation changes a lot when using various model and latent space sizes and found that the correlation remains high across all models, ranging from 0.87 to 0.93 (see Supplementary Table 3).

We then assessed the structural properties of the variants by predicting their three dimensional structure using ESMFOLD, performing inverse folding sequence scoring using the wildtype *Ec*NAGK structure and then downstream dynamics analyses using ensemble Normal Mode Analysis (eNMA). The predicted structures have an average pLDDT of 89.64 (see Fig. 5A), which confirms high accuracy of the predictions, closely resembling the structure of the wildtype enzyme (average RMSD: 1.70Å). This result was further confirmed when scoring our variants against the wildtype *Ec*NAGK structure, where the average inverse folding score of -0.077 (see Fig. 5B, C), suggesting that variants are highly consistent with the wildtype fold albeit to a lesser extent than the wildtype sequence (wildtype log-likelihood: -0.991; average variants log-likelihood: -1.069). We then looked at the dynamics of each variant and, in general, we found strong agreement with the wildtype dynamics (average RMSF: 0.15, (see Fig. 6A)). Nonetheless, we found that the region spanning residues 58 and 65, and 211 and 216, showed increased flexibility, which could potentially lead to phenotypic differences (see Fig. 6B).

Taken together, the overall dynamics of the protein is preserved despite the introduced mutations.

### Establishing the auxotrophic screen for *Ec*NAGK

The *Ec*NAGK encoded by the *argB* gene is conditionally essential for protein synthesis in the absence of exogenous L-arginine (or a suitable precursor). In this context, we hypothesised that *argB* knockouts could be rescued by expressing *Ec*NAGK (or functional variants) via an auxiliary plasmid, which is supplemented by DNA transformation. Without access to such a plasmid, we reasoned that the auxotroph will not survive in L-arginine-deficient media. Therefore, by using a simple viability screen, it becomes possible to identify functional *Ec*NAGK variants in a manner amenable to high-throughput screening.

To test this hypothesis, we cured (see Supplementary Fig. 4) and transformed commercial BW25113 Δ*argB* using a bespoke constitutive expression vector encoding the wildtype *E. coli argB* gene (pKCHU-argB) and the production of *Ec*NAGK was subsequently confirmed by SDS-PAGE (see Supplementary Fig. 5A–C). As anticipated, L-arginine auxotrophy could be remedied in M9 minimal salts media by either the supplementation of L-arginine or by complementation with pKCHU-argB (see Supplementary Fig. 5D). However, using this plasmid-based expression system, we observed that *argB* knockouts required over 24 h of incubation to emerge from lag phase. Based on our SDS-PAGE analysis, we hypothesised that the strong *Ec*NAGK overexpression was severely penalising cell growth in minimal media. By recoding the *argB* coding sequence using *E. coli* codon usage bias data (see Supplementary Fig. 6)[16], we measured an average 25-fold reduction in gene expression by RT-qPCR, which was associated with a drastic improvement in growth rate using both solid and liquid minimal media (see Supplementary Fig. 7). Given the clear benefit to cell viability, we decided to backtranslate all our variants using the same codon usage bias strategy.

### Screening the *Ec*NAGK variant library

A sequence-perfect library of 40 *Ec*NAGK variants (Neochromosome, Inc.) cloned into our pKCHU expression vector was used to transform BW25113 Δ*argB*. To this end, we developed a robust and high-throughput heat-shock transformation protocol using an Opentrons OT-2 robot, which allowed us to reliably and reproducibly transform *E. coli* using both pET23b-EFGP and pKCHU-argB with minimal user input (see Supplementary Fig. 8). Whilst many *Ec*NAGK variants yielded colonies overnight following heat-shock transformation, several more variants exhibited poorer transformation efficiency and required up to 2 consecutive nights of incubation to develop colonies (see Supplementary Fig. 9). Despite numerous attempts, 7 *Ec*NAGK variants (namely poe4, snl3, snl6, snl7, sni10, sne1, sne3) did not produce viable transformants. The observed array of cell viability suggests that the *Ec*NAGK variants are not tolerated equally, and some variants may considerably affect cell fitness.

The remaining 33 viable variants were studied by auxotrophic selection, as previously outlined. Remarkably, 22 of the 33 viable *Ec*NAGK variants could quickly recover the BW25113 Δ*argB* phenotype in standard M9 minimal salts solid media without L-arginine supplementation. Interestingly, these variants include the top-ranked viable candidates in each model category (poe1, snl1, sni1, sne2). Further 6 variants (poe5, sni9, sne5, sne7, sne8, sne9) facilitated slow-to-intermediate recovery, thus bringing the total count of active variants to 28. The remainder (poe2, poe3, poe8, snl5, snl10) did not

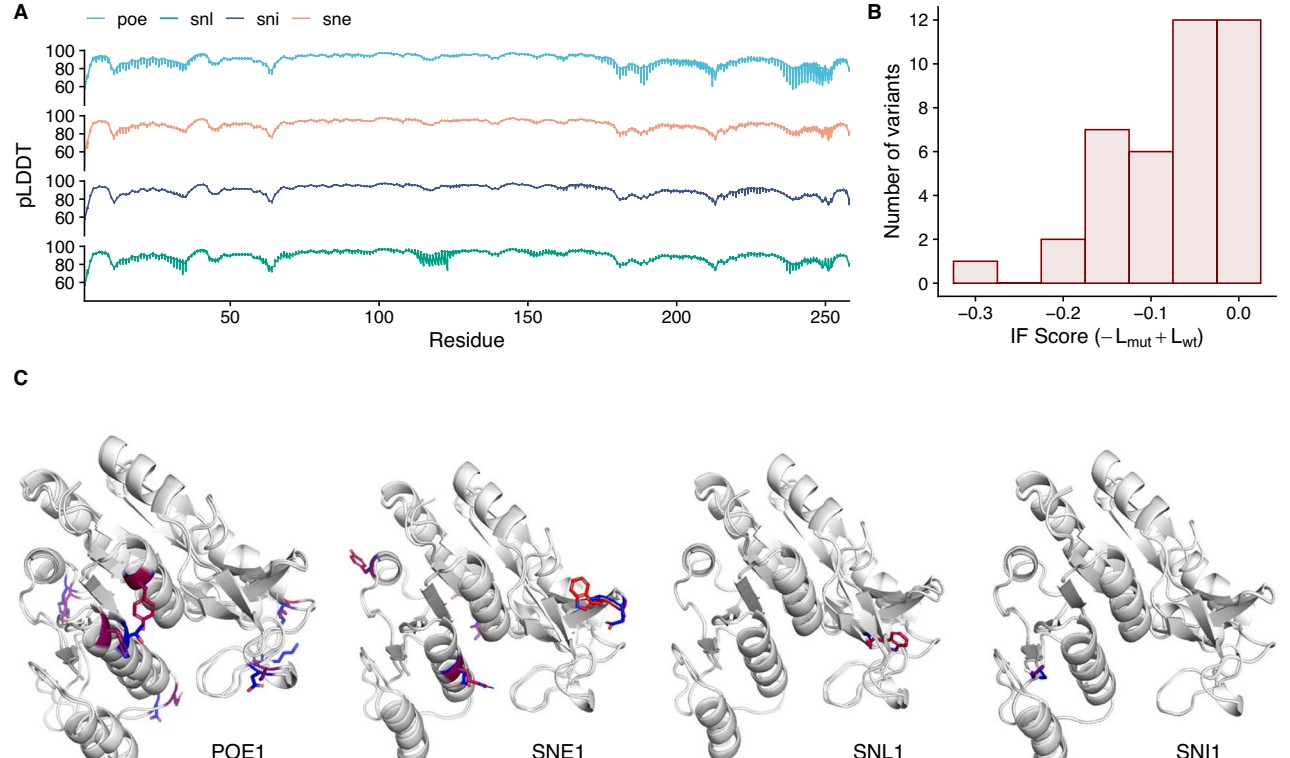

**Fig. 5 | Structural analysis of the *Ec*NAGK variants library designed by PRE-VENT. A** Accuracy of the *Ec*NAGK variants structures predicted by ESMFOLD reported as predicted local distance difference test (pLDDT) at the residue level; regions with pLDDT > 70 are expected to be accurately modelled. **B** Inverse folding scores of the variants library. **C** Structure of the top variant for each design strategy, with mutations color coded based on the BLOSUM90 substitution matrix scores.

rescue the phenotype after 48 hours. Conversely, all cells carrying our variants could proliferate readily and rapidly by supplementing the minimal media with L-arginine (see Fig. 7A).

We then checked whether using the predicted energy information for designing or prioritizing variants was associated with either a higher number of functional variants or more diverse sequences. 14 out of the 28 viable variants were obtained by using either POE (6 variants) or SNE (8 variants); as expected, since they carry only 1 mutation, 9 out of 10 SNI variants recovered the phenotype, whereas only 5 out of the 10 SNL variants were functional.

It is interesting to note that prioritizing variants by ELBO did yield the lowest number of viable variants, suggesting that it might not be the most effective strategy. Interestingly, functional variants designed using free energy information (POE, SNE) harbour, on average, significantly more mutations (5.64) compared to the 1.64 mutations found on variants selected by sequence only information (Student t-test; t: 4.9173, df: 22.177, *p*-value = $3.151 \times 10^{-5}$), suggesting that exploiting free energy information could yield more diverse library with minimal impact on the overall number of functional proteins obtained in a screen.

We subsequently quantified the growth supported by the top candidates for each design strategy (namely poe1, snl1, sni1, and sne2) in a cell density time-course assay. We observed that variants poe1, sni1, and sne2 enabled similar rates of exponential growth compared to the wildtype *Ec*NAGK (see Fig. 7B, C), albeit cells expressing sne2 exhibited a 23−34% shorter lag time compared to all other experiments (see Supplementary Table 7). Conversely, cells expressing variant snl1 doubled 39% slower on average than those expressing the wildtype enzyme.

We then analyzed whether the introduced mutations in the top candidates are located at more buried or more accessible sites in the protein structure. We computed the smoothed hydropathy index[17] for each residue in the wildtype and checked the location of the mutations

in the top candidates. Around 33% of the mutations are located in the most hydrophobic regions of the protein, which suggests that the mutations are not biased towards the surface of the protein (see Supplementary Fig. 10).

Notably, variant snl1 features a radical and sterically-intrusive Gly123Trp mutation in an otherwise highly-conserved position on β7, which likely destabilizes the β6-β7 hairpin in the NAG-binding subdomain (see Supplementary Fig. 11)[18]. Furthermore, the expression of variant snl1 was discovered to be 5-fold greater than the wildtype by RT-qPCR analysis (see Supplementary Fig. 12). Therefore, we propose that the observed growth penalty may be attributed to the divergent nature of the Gly123Trp mutation and/or less favorable transcription regulation, which is mediated by our gene re-coding strategy.

In conclusion, 85% of the designed and transformed *Ec*NAGK variants enabled survival on auxotrophic media, with 67% permitting similar recovery levels to the wild-type sequence.

## Discussion

Computational protein engineering has been limited by the complexity and poor understanding of the function driving the folding of a polypeptide chain into a thermodynamically stable structure.

Here we addressed these problems by building a generative deep learning model, called PREVENT, which uses variational inference to approximate the free energy landscape of a target protein by learning favorable mutations from a small number of biophysical simulations. Our method enables generation of new protein sequences and simultaneous prediction of the associated free energy and, more importantly, enables free energy minimization over a mathematically tractable thermodynamic space, which enables the rapid identification of diverse putatively functional variants. Importantly, PREVENT enables large scale protein re-engineering, since the cost of predicting the free energy function remains constant with respect to the number

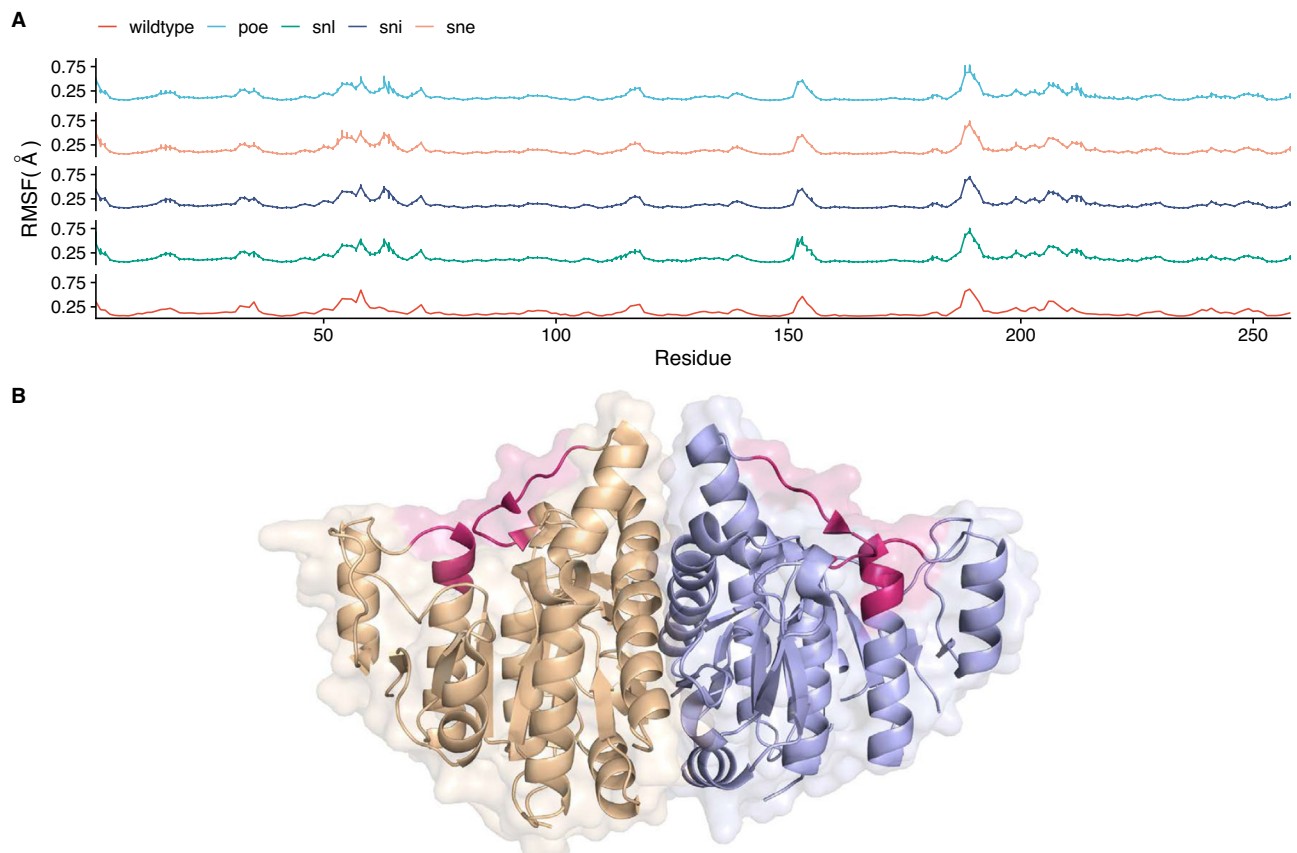

**Fig. 6 | ensemble Normal Mode Analysis (eNMA) of the *Ec*NAGK variants library designed by PREVENT. A** Residue level Root Mean Square Fluctuation (RMSF) of *Ec*NAGK variants compared to wildtype. The most significant changes with respect to the wildtype are in the regions comprising residues 58 and 65, and 211 and 216. **B** Dimer structure of *Ec*NAGK with region of increased flexibility compared to the wildtype colored in red.

of mutations, whereas even simple random search using directly FOLDX is computationally intractable beyond a handful of mutations.

We used our model to redesign the *E. coli* phosphotransferase N-acetyl-L-glutamate kinase (*Ec*NAGK), a key enzyme of the L-arginine pathway characterized by a highly dynamic structure, which is likely to be associated with multiple stable thermodynamic states. Computational results showed that PREVENT can accurately approximate the free energy landscape of this protein and generate new and diverse variants, which retain wildtype structural features, irrespective of the size of the training set used. We then built a library of *Ec*NAGK variants and experimentally shown that 85% of the generated variants were functional, albeit not all variants could completely recover the wildtype growth phenotype. Importantly, design or selection of variants using predicted free energy information allowed us to obtain a significantly more diverse set of functional proteins compared to sequence only selection criteria, further validating our approach. This is particularly interesting for drug discovery, especially with the advent of noninferiority trials to assess bioequivalence of a new formulation to an established drug[19]; in fact, drugs can be approved even if marginally inferior, when other aspects of the treatment, such as its cost, are important. A method like PREVENT can significantly speedup the engineering of these bioequivalent molecules.

While PREVENT allowed redesigning a complex enzyme, we are also aware of its limitations. Currently, our free energy estimates are obtained by FOLDX, a well-known and validated tool, which has been shown to be reliable mostly for small and medium size proteins, mostly due to the intrinsic difficulty of modeling proteins in unfolded state[20]. Understanding how the accuracy of the free energy calculations will impact our model will require further analysis using different biophysical engines, including neural network potentials[21], and proteins

from different families. Moreover, while the use of free energy information for protein engineering is reasonable, the limited accuracy of current estimation methods suggests that relative changes, $\Delta\Delta G$, in free energy compared to the wildtype protein could be a more robust and accurate way to exploit energy information, especially for large proteins. Finally, we also recognize that models like PREVENT could help unveil general design rules associated with stabilizing mutations, albeit identifying the best framework to make our model explainable will require further investigation.

Taken together, our work provides a new framework to integrate generative deep learning with biophysical information, which can substantially improve both the number and diversity of functional proteins obtained compared to more lengthy and expensive molecular biology approaches. It is also important to note that while PREVENT learns the sequence-to-free-energy relationship of a protein, it can learn relationships with any other biochemical property, thus representing a generic framework for phenotype-driven protein design.

## Methods

### In silico mutagenesis and free energy estimation

In order to approximate the *E. coli* phosphotransferase N-acetyl-L-glutamate kinase (*Ec*NAGK) free energy landscape, we generated variants by mutating the wildtype protein uniformly at random, except for the methionine in the first position, and allowing up to 15% of the residues to be mutated, while discarding any duplicated sequences.

We then estimated Gibbs free energy of each variant using the FoldX empirical force field, which has been shown to be a robust tool to estimate the stability of protein and protein complexes[14,20]. Specifically, we used standard FOLDX best practice[14], where we first

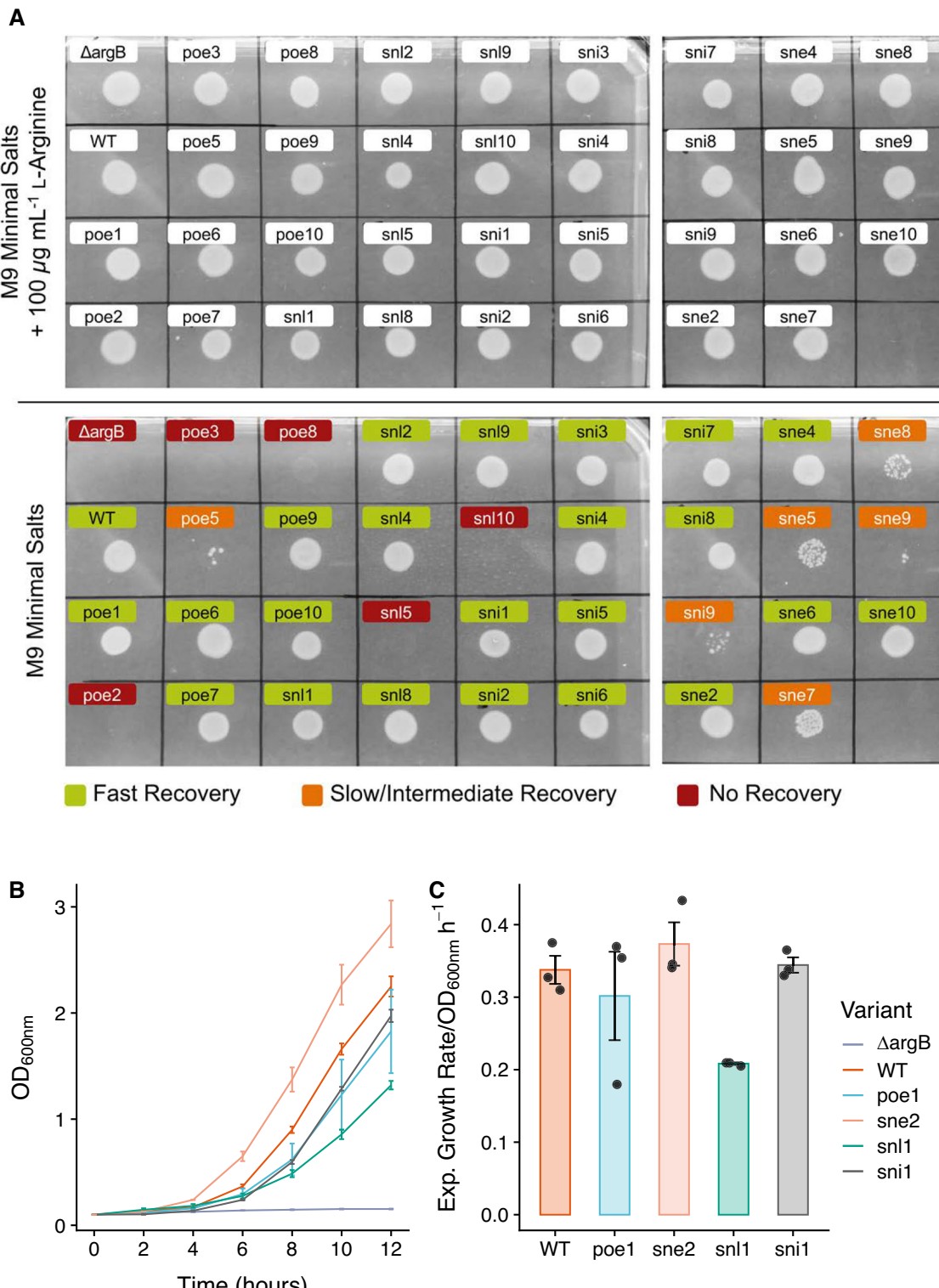

**Fig. 7 | Functional analysis of the *Ec*NAGK variants designed by PREVENT. A** Top plate: perfect recovery of transformed *Ec*NAGK variants, including wildtype and Δ*argB*, in L-arginine rich media. Bottom plate: different recovery rates of transformed *Ec*NAGK variants in the minimal media. **B** Average growth curves of BW25113 (*argB*) and *Ec*NAGK transformants grown in auxotrophic M9 minimal salts media. Optical density (OD) measurements were recorded every 2 h for up to 12 h. Error bars represent the standard error of three biological replicates. **C** The average rate of exponential growth supported by wildtype (*Ec*NAGK) and variants. Error bars represent the standard error of three biological replicates.

performed structural relaxation of the wildtype structure, using the REPAIRPDB command, which was then used to estimate the Gibbs free energy ($\Delta G$) of the variants using the BUILDMODEL command using default parameters and averaging the value across 2 runs.

**Joint variational inference of protein sequences and free energy**
We hypothesised that a generative model could approximate the sequence-to-free-energy function by fitting a latent, multivariate statistical distribution, whose parameters are unknown but can be

estimated from the data. Specifically, we assume that samples of the latent distributions encode both sequence and free energy information, which allows us to explore the protein design space either with respect to the sequence properties or to free energy states.

To do that, we built a new generative protein engineering model, called PRotein Engineering by Variational frEe eNergy approximaTion (PREVENT), by extending the classical Variational Autoencoder (VAE) framework[11], which has been shown to be effective in designing new, functional proteins[6,22,23]. In a classical VAE model, the objective is to maximize the Evidence Lower BOund (ELBO):

$$\mathcal{L}(\boldsymbol{\phi}, \boldsymbol{\theta}) = \mathbb{E}_{q_{\boldsymbol{\phi}}(\boldsymbol{z}|\boldsymbol{x})}[\log p_{\boldsymbol{\theta}}(\boldsymbol{x}|\boldsymbol{z})] - KL(q_{\boldsymbol{\phi}}(\boldsymbol{z}|\boldsymbol{x})||p_{\boldsymbol{\theta}}(\boldsymbol{z})) \rightarrow \max_{\boldsymbol{\theta}, \boldsymbol{\phi}} \quad (1)$$

where $\boldsymbol{x}$ is an observed random variable (i.e. a protein sequence) and $\boldsymbol{z}$ is latent variable (or encoding) and KL is the Kullback-Leibler divergence. Our model, instead, takes into account two observed variables, that is a sequence $\boldsymbol{x}$ and its associated $\Delta G$ value $g$, which we assume to be conditioned on the latent variable $\boldsymbol{z}$. Since $\boldsymbol{x}$ and $g$ are conditionally independent given $\boldsymbol{z}$, the entire probability distribution is factorized as $p(\boldsymbol{x}, g, \boldsymbol{z}) = p(\boldsymbol{z})p(\boldsymbol{x}|\boldsymbol{z})p(g|\boldsymbol{z})$. By assuming that the variational distribution $q_{\boldsymbol{\phi}}(\boldsymbol{z}|\boldsymbol{x}, g) = q_{\boldsymbol{\phi}}(\boldsymbol{z}|\boldsymbol{x})$, since we only model the latent variable dependency on the sequence information, our model can be trained end-to-end by maximizing the ELBO defined as follows:

$$\mathcal{L}(\boldsymbol{\phi}, \boldsymbol{\theta_1}, \boldsymbol{\theta_2}) = \mathbb{E}_{q_{\boldsymbol{\phi}}(\boldsymbol{z}|\boldsymbol{x})}[\log p_{\boldsymbol{\theta_1}}(\boldsymbol{x}|\boldsymbol{z})] + \mathbb{E}_{q_{\boldsymbol{\phi}}(\boldsymbol{z}|\boldsymbol{x})}[\log p_{\boldsymbol{\theta_2}}(g|\boldsymbol{z})]$$
$$- KL(q_{\boldsymbol{\phi}}(\boldsymbol{z}|\boldsymbol{x})||p(\boldsymbol{z})) \rightarrow \max_{\boldsymbol{\theta_1}, \boldsymbol{\theta_2}, \boldsymbol{\phi}} \quad (2)$$

where $\mathbb{E}_{q_{\boldsymbol{\phi}}(\boldsymbol{z}|\boldsymbol{x})}[\log p_{\boldsymbol{\theta_2}}(g|\boldsymbol{z})]$ corresponds to the reconstruction error for $\Delta G$. Specifically, by assuming that $p(g|\boldsymbol{z}) = N(g|\mu_{\boldsymbol{\theta_2}}(\boldsymbol{z}), 1/\sqrt{2})$, this term is proportional to $\mathbb{E}_{q_{\boldsymbol{\phi}}(\boldsymbol{z}|\boldsymbol{x})}[-(g - \mu_{\boldsymbol{\theta_2}}(\boldsymbol{z}))^2]$, which is equivalent to the Mean Squared Error (MSE).

To approximate $p_{\boldsymbol{\theta_1}}(\boldsymbol{x}|\boldsymbol{z})$ and $q_{\boldsymbol{\phi}}(\boldsymbol{z}|\boldsymbol{x})$, we used a standard Transformer architecture, a widely used neural network architecture for language processing[24]; here we hypothesized that an attention mechanism could better assist in the prediction of $\Delta G$ values from a variant than any other architecture, given the high sequence homology in the training set. PREVENT uses 6 layers for the transformer encoder and 4 layers for the decoder with 512 embedding size split across 8 heads. To approximate $p_{\boldsymbol{\theta_2}}(g|\boldsymbol{z})$, instead, we used a Fully connected Neural Network (FCNN) with 5 layers of progressively decreasing size (from 64 to 1 nodes) and RELU as non-linear activation function, except for the linear transformation in the last layer.

Finally, since the variance of the free energy can be extremely large between batches and higher free energy values are more likely to appear as most mutations are destabilizing, we developed a weighted random sampler to ensure a uniform coverage of the sequence and free energy space during training. Specifically, a histogram of free energy values was first constructed and then sequences were included in a mini-batch probabilistically at random, with a probability inversely proportional to the number of sequences in their corresponding free energy bin.

## Protein engineering using variational energy approximation

Our generative model allows to learn a robust approximation of the sequence-to-free-energy function, which can then be exploited to generate more stable proteins. Specifically, PREVENT implements two free energy based design procedures, namely the seeded non optimal energy ranking (SNE) and the prior optimization of free energy (POE).

The SNE procedure follows the classical VAE approach, where a seed sequence is passed as input to the encoder to obtain an embedding $\boldsymbol{z}$, which in turn is passed in input both to the sequence decoder, to obtain an amino acid sequence, and the energy decoder to estimate the predicted free energy. Specifically, while sequences can be

generated using categorical sampling, the expected free energy estimate associated with each generated sequence, instead, is obtained by first passing a generated sequence $\boldsymbol{x}$ in input to the encoder to obtain $q(\boldsymbol{z}|\boldsymbol{x})$, and then the resulting encoding to the free energy decoder to obtain $\mathbb{E}[g|\boldsymbol{x}]$, that is the expected value of the energy given $\boldsymbol{x}$, as shown in Eq. (3). In practice, we estimate the variational posterior free energy distribution of $\mathbb{E}[g|\boldsymbol{x}]$ as follows:

$$\mathbb{E}[g|\boldsymbol{x}] = \mathbb{E}_{q(\boldsymbol{z}|\boldsymbol{x})}|\mathbb{E}_{p(g|\boldsymbol{z})}[g]| = \mathbb{E}_{q(\boldsymbol{z}|\boldsymbol{x})}[\mu_{\boldsymbol{\theta_2}}(\boldsymbol{z})] \approx \frac{1}{N}\sum_{n=1}^{N} \mu_{\boldsymbol{\theta_2}}(\tilde{\boldsymbol{z_n}}) \ \tilde{\boldsymbol{z_n}} \sim q(\boldsymbol{z}|\boldsymbol{x})$$

(3)

where $N = 300$ samples in our experiments.

While the energy ranking strategy enables the prioritization of variants based on their predicted free energy, it does not inform the sequence generation process. However, since PREVENT also maps free energy information to the latent space, we can perform free energy optimisation over the latent space and then use the resulting encoding to sample new sequences; specifically, we hypothesized that this approach would allow us to more efficiently navigate the protein design space and to generate more diverse proteins. To do that, we used a constrained trust region method[25] to minimize the predicted free energy $f_{\boldsymbol{\theta_2}}(\boldsymbol{z})$, while constraining $\boldsymbol{z}$ within 3 standard deviations from the mean to control sequence diversity. Once an embedding $\boldsymbol{z_{opt}}$ associated with a minimal free energy is found, it is used as input to the sequence decoder to generate new variants, which are then ranked by their expected energy values.

## Computational structural analysis of generated proteins

To characterise the structural and dynamic properties of the generated variants, we developed a standardized analytical workflow.

First, we obtained structure prediction for each variant using ESMFOLD[26] with default parameters and then performed potential energy minimization to relax the predicted structure as implemented in the OPENMM package[27] using the Amber14 force field, while adding an harmonic potential energy term to restrain Cα atoms position. ensemble Normal Mode Analysis (eNMA) was performed using the BIO3D package[28], using the `calpha` force field and by setting the temperature to T = 300K. Downstream fluctuation analysis was conducted by generating trajectory using the 7th mode. Finally, we evaluated the likelihood of the variants to be compatible with the known wildtype structure. To do that, we used the ESM inverse folding model (ESM-IF)[29] using the wildtype (*Ec*NAGK) structure, and scoring variants using the score $IF(\boldsymbol{s_{mut}}, \boldsymbol{s_{wt}}) = \mathcal{L}(\boldsymbol{s_{mut}}) - \mathcal{L}(\boldsymbol{s_{wt}})$, where $\mathcal{L}$ is the log likelihood of a sequence given the input structure. IF values greater or equal to 0 are indicative of highly structurally compatible variants.

## Experimental materials

BW25113 Δ*argB* Keio knockout strain was purchased from Horizon Discovery Ltd. Basic parts for plasmid creation were kindly supplied by Dr. Marcos Valenzuela-Ortega. Polymerases, RNA miniprep kits, restriction enzymes and master mixes for cloning were purchased from New England Biolabs. Invitrogen E-Gel (1%, with SYBR Safe) agarose gels were used for DNA electrophoresis. Invitrogen NuPAGE (4−12% Bis-Tris) gels and 1X MES NuPAGE SDS Running buffer were used for SDS-PAGE (see Supplementary Fig. 14). 5X M9 minimal salts base was purchased from Formedium. All other chemical reagents were purchased from Merck or Fisher Scientific. The *Ec*NAGK variant library was assembled and prepared by Neochromosome Inc. The High-throughput transformations and agar plate spotting were achieved using an Opentrons OT-2 robot equipped with a thermocycler module, a single channel p20 pipette and a multi-channel p300 pipette. Where stated, the working concentration of kanamycin or carbenicillin in selective media was $50 \mu$ g mL$^{-1}$ and $100 \mu$ g $m L^{-1}$, respectively.

## Construction of the pKCHU-argB vector

pKCHU-argB was assembled via the Joint Universal Modular Plasmids (JUMP) method using pre-domesticated parts and destination vector (see Supplementary Table 4)[30]. In brief, the basic parts (2 $fmol\,\mu L^{-1}$ each) and destination vector (2 $fmol\,\mu L^{-1}$) were combined with BsaI (1U) and NEBuilder Master Mix (1X) in a total reaction volume of 15 $\mu L$. The parts were assembled using the reported thermocycler parameters for a level 0 JUMP assembly[31]. The reaction mixture was used to transform commercial NEB5-alpha cells (New England Biolabs) using the heat-shock method. Transformants were selected overnight at 37 °C on Yeast Extract-Peptone (YEP) agar plates supplemented with kanamycin. Potential constructs were identified by colony PCR (see Supplementary Fig. 13), propagated in YEP-kanamycin medium (10 mL), isolated using a GeneJET Plasmid Miniprep kit (Thermo) and confirmed by Sanger sequencing.

## Purification of EcNAGK variant DNA library

The EcNAGK variant DNA library was assembled and prepared commercially using the same basic parts as pKCHU-argB. Dried E. coli Top 10 cells, harbouring sequence-perfect clones, were individually reconstituted in yeast-extract-peptone (YEP) medium (200 $\mu L$) at 37 °C for 6 h. The reconstituted cells were propagated in fresh YEP medium (10 mL) for a further 18 hours at 37 °C. Following biomass harvest by centrifugation (4000 rcf, 15 minutes), the plasmids encoding the (EcNAGK) variants were isolated using a GeneJET Plasmid Miniprep kit (Thermo). The plasmids were stored at −20 °C in Tris-HCl buffer (0.1 M, pH 8) until required.

## Curing, preparation and transformation of chemically-competent E. coli BW25113 ΔargB

Commercial E. coli BW25113 ΔargB was cured by Flp-FRT recombination using the curing plasmid pCP20, as per literature protocol[32]. The loss of both kanR and ampR from BW25113 ΔargB was confirmed by the restored susceptibility to kanamycin and carbenicillin. Cured BW25113 ΔargB was sub-cultured in YEP media (20 mL, 37 °C) until mid-log phase. The biomass was harvested by centrifugation (3000 rcf, 10 minutes, 4C) and washed thrice with sterile, ice-cold CaCl2 (100 mM, 20 mL). The washed cells were resuspended in ice-cold $CaCl_2$ (100 mM, 2 mL), sub-aliquoted as required and transformed fresh using the heat-shock method. pKCHU-argB transformants were selected on YEP-agar (37 °C, 18 h) laced with kanamycin.

## Auxotrophic screen on solid minimal media

BW25113 ΔargB transformants (harbouring pKCHU-argB or variants thereof) were propagated in YEP-kanamycin medium (10 mL) overnight with agitation (37 °C, 18 h). Culture samples ($4.8 \times 10^8$ cells) were diluted in PBS (500 $\mu L$, 0.1 mM, pH 7.4), gently harvested and washed a further two times in PBS (200 $\mu L$) to remove residual rich media. The washed cells were resuspended in PBS (100 $\mu L$) and spotted (5 $\mu L$) on solid minimal media containing bacto-agar (1% w/v), M9 minimal salts base (1X), $MgSO_4$ (2 mM), $CaCl_2$ (0.1 mM) and D-glucose (0.4% w/v). Positive control media was prepared by supplementing with L-arginine (100 $\mu gmL^{-1}$). The minimal media plates were incubated (37 °C) and visually inspected for growth over 48 h.

## Auxotrophic selection on liquid minimal media

BW25113 ΔargB transformants (harbouring pKCHU-argB or variants thereof) were propagated in YEP-kanamycin medium (10 mL) overnight with agitation (37 °C, 18 h). The washed cells were resuspended in PBS (10 mL) and sampled for back-dilution in minimal media (25 mL, OD600 = 0.1) containing M9 minimal salts base (1X), $MgSO_4$ (2 mM), CaCl2 (0.1 mM) and D-glucose (0.4% w/v). The cultures were incubated (37 °C) with vigorous shaking for 12–48 h. $OD_{600}$ measurements were recorded periodically.

## Reverse Transcriptase Quantitative PCR (RT-qPCR) of wildtype constructs

BW25113 ΔargB transformants (harbouring pKCHU-argB) were propagated in YEP-kanamycin medium (10 mL) overnight with agitation (37 °C, 18 h). Untransformed BW25113 ΔargB was similarly propagated in antibiotic-free YEP. The cells were sub-cultured ($OD_{600} = 0.1$) in fresh YEP media and harvested by centrifugation (3000 rcf, 12 minutes) at mid-log phase. The cells were resuspended in PBS (1 mL) and pelleted again by microcentrifugation (3000 rcf, 5 min). Total RNA was extracted from the pelleted biomass using a Monarch Total RNA Miniprep kit (New England Biolabs). Per RT-qPCR reaction (20 $\mu L$), cDNA was synthesised and amplified from total RNA (500 ng) with target-specific primers (400 nM) using a Luna Universal One-Step RT-qPCR kit (New England Biolabs), according to manufacturer protocol. The reactions were performed and analysed in a StepOnePlus RT-qPCR system (Thermo) configured for SYBR Green fluorescence detection. Reverse transcriptase-free and template RNA-free controls were performed in parallel. Relative fold-expression was calculated via the Livak method using E. coli 16S rRNA (rrsA) as the reference gene.

## Reporting summary

Further information on research design is available in the Nature Portfolio Reporting Summary linked to this article.

## Data availability

The data used and generated in this study have been deposited in Zenodo at under https://doi.org/10.5281/zenodo.13763481. Details for reproducing the figures, plots and tables are provided in the source data file. Source data are provided in this paper.

## Code availability

The software is available at the following https://github.com/stracquadaniolab/prevent-nf[33].

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

## Acknowledgements

This work was supported by the UKRI EPSRC Fellowship (EP/V033794/1), the UKRI EPSRC grant (EP/Y01913X/1) and the IBioIC feasibility funding (FF-2022-04) to G.S. The UKRI Centre for Doctoral Training in Biomedical AI (grant EP/S02431X/1) for E.L. and the UKRI Biotechnology and Biological Sciences Research Council (BBSRC) grant number BB/T00875X/1 for M.K. Computational experiments were performed using resources provided by the Cambridge Service for Data Driven Discovery (CSD3) operated by the University of Cambridge Research Computing Service (www.csd3.cam.ac.uk), provided by Dell EMC and Intel using Tier-2 funding from the Engineering and Physical Sciences Research Council (capital grant EP/P020259/1), and DiRAC funding from the Science and Technology Facilities Council (www.dirac.ac.uk).

## Author contributions

G.S. conceived the study. G.S. and E.L. formulated and developed the PREVENT model. E.L. implemented and tested the model and performed sequence analysis. M.K. developed a transformation protocol for the Opentrons OT-2 robot. M.H. and M.K. performed lab experiments. G.S. supervised all experimental work. G.S., E.L., and M.H. wrote the manuscript.

## Competing interests

Giovanni Stracquadanio is Co-Founder of ZYTHERA, a startup of the University of Edinburgh using AI and engineering biology to develop enzyme replacement therapies. The other authors declare no competing interests.
