## [Transparent Peer Review file · Nature Communications]

Protein engineering using variational free energy approximation

Corresponding Author: Professor Giovanni Stracquadanio

Version 0:

Reviewer comments:

Reviewer #1

(Remarks to the Author)

The manuscript by Stracquadanio and colleagues describes their protein engineering work targeting a specific protein using a generative model for protein sequences. An encouraging aspect of their work is the thorough wet-lab validation conducted. In contrast to numerous recent publications lacking experimental validation, this study offers evidence of the model's efficacy in real-world applications, which deserves recommendation. However, I have several major concerns regarding the computational aspects that need addressing prior to publication:

1) Significance of introduced mutations: Since the overall enzyme activity wasn't notably enhanced compared to the wild type (as I understand), I suspect mutations are possibly located at more accessible sites (e.g., surface polar residues). If this is the case, the necessity of a deep learning model might be questioned.

2) Code availability & reproducibility: The absence of a repository for the training code. Given that their model is currently trained for EcNAGK, and applying it to other proteins necessitates a new training procedure, providing the training code along with feature extraction code is important to gather interest within the scientific community.

3) Computational details: It's essential to provide specifics regarding the computational procedures. How long does it take to generate FoldX variants for the entire dataset or a subset of 25k data? Additionally, what's the duration of model training? Furthermore, critical aspects such as the choice of hyperparams (such as latent dimension) will be worth reporting. Especially the latent dimension of 128 seems relatively large considering the dataset size and the type of the problem (reconstruction may improve but sampling may become much noisier and harder) – I suggest ablation study to validate their hyperparameter choices.

Reviewer #2

(Remarks to the Author)

****Summary****

In this study, the authors introduce a novel deep learning method, using a Variational Autoencoder (VAE), to approximate the free-energy landscape of proteins. This method, termed PREVENT (PRotein Engineering by Variational frEe eNergy approximaTion), aims to enhance protein thermostability and functionality by generating new, stable, and functional protein sequences. The authors conducted experiments on E. coli phosphotransferase N-acetyl-L-glutamate kinase (EcNAGK) to demonstrate how this free-energy-associated generative model can be beneficial in protein engineering.

****Strengths****

1. The concept of employing VAE to approximate the joint distribution between protein sequences and free energy is well-articulated. VAEs, known for their efficiency in deep learning without requiring an explicit likelihood function (which can be computationally demanding), facilitates the learning of interpretable latent variables, proving to be advantageous for protein engineering tasks.

****Weaknesses****

1. It is unclear whether the PREVENT-induced mutations are superior to the variants present in the training set. Figure 1.A indicates that the training data's free energy spans from -40 to 150 kcal/mol, while Figure 4.A shows that the generated sequences' free energy peaks at around -30 kcal/mol. If the free energy of generated sequences is not more favorable than that of random mutations, one might question the advantage of PREVENT over extensive random mutation paired with a tool like FOLDX for energy prediction and selection of the highest-energy variants.

2. The correlation between free energy and functional performance, as tested experimentally, needs clarification. Since the design of sequences heavily relies on predicted free energy, it is crucial to validate if lower energy correlates with enhanced functionality.

3. The training process details are vague, particularly regarding how the training set was divided and what criteria were used to terminate the training. With 1,400 epochs mentioned, there's a concern that the model might have overfitted the dataset.

4. In the EcNAGK experiment section, the authors mention generating 117,387 unique variants, but the method for generating these variants remains unclear. The method section suggests that variants were randomly generated; it should be specified if this applies to EcNAGK as well and the time required to label these variants with free energy.

Errors

- Line 306: The symbol 'phi' should be subscripted.

- Line 329: The phrase "seed sequence is passed in input to the decoder" should be corrected to "seed sequence is passed as input to the encoder," as it involves encoding the input sequence to obtain an embedding, not decoding.

Reviewer #3

(Remarks to the Author)

Great paper, clearly written and easy to digest despite the complexity of the work done including the design of the computational method, substantial in silico experiments as well as wet lab experimental validation.

In terms of writing clarify I have only a few comments:

- Can the authors describe better the held-out test set used to evaluate the performance of the generator across the four training set sizes? What size it had, how where the sequence in it selected?

- It would be helpful to present a graphical description of the proposed architecture to help understand how all components fit together.

- "and KL divergence is the Kullback-Leibler." -> "and KL is the Kullback-Leibler divergence."

Finally, I think that there is a missing opportunity in this work which is to dig into the decision making mechanisms of the networks in order to distill (if possible) biological knowledge relevant for the sequence generation and free energy prediction. The use of transformers (with their attention mechanisms) should facilitate this process.

Version 1:

Reviewer comments:

Reviewer #1

(Remarks to the Author)

The authors addressed all my concerns. I have no further concerns.

Reviewer #2

(Remarks to the Author)

The authors have solved my concerns in the previous review.

Reviewer #3

(Remarks to the Author)

It is very difficult to assess the revision of the manuscript because I cannot see a revised manuscript with track of changes, and the authors's reply to my comments is minimal or literally non-existent. I'm pasting from the reply letter, with my comments [in between brackets].

- Can the authors describe better the held-out test set used to evaluate the performance of the generator across the four training set sizes? What size it had, how where the sequence in it selected?

We thank the reviewer for the comments, and we have now further clarified our procedures.

[It would have been nice to have further details in the reply letter, given the lack of track of changes in the revised

manuscript]

- It would be helpful to present a graphical description of the proposed architecture to help understand how all components fit together.

[There is no reply whatsoever to this comment]

- "and KL divergence is the Kullback-Leibler." -> "and KL is the Kullback-Leibler divergence."
Fixed.

[It is not fixed. Page 11, line 331].

Finally, I think that there is a missing opportunity in this work which is to dig into the decision making mechanisms of the networks in order to distill (if possible) biological knowledge relevant for the sequence generation and free energy prediction. The use of transformers (with their attention mechanisms) should facilitate this process.

We thank the reviewer for the suggestion but, at this time, we think this is beyond the scope of the paper.

[It would have been nice to get a more elaborated answer, at least explaining why it is not feasible in the short term to do this analysis. The reliability of AI models is a very important matter, and this analysis can greatly contribute to it]

In these conditions it is very difficult to judge whether these revisions are sufficient to accept the manuscript.

Version 2:

Reviewer comments:

Reviewer #3

(Remarks to the Author)

The authors have addressed all of my concerns.

Response to reviewers

Reviewer #1 (Remarks to the Author)

1) Significance of introduced mutations: Since the overall enzyme activity wasn't notably enhanced compared to the wild type (as I understand), I suspect mutations are possibly located at more accessible sites (e.g., surface polar residues). If this is the case, the necessity of a deep learning model might be questioned.

We thank the reviewer for the useful comments, which helped make our analysis stronger. Specifically, sne2 variant grows twice as fast as the wildtype (or any other variant), suggesting that a gain of function is obtained by our approach, albeit we agree that further analysis of the kinetics of the enzyme would be required to make a conclusive statement. However, we think this level of validation is beyond the scope of this work. Regarding the location of our mutations, we briefly discussed it at lines (252-257), and showed that the mutations in the top candidates are not significantly biased towards accessible sites.

2) Code availability & reproducibility: The absence of a repository for the training code. Given that their model is currently trained for EcNAGK, and applying it to other proteins necessitates a new training procedure, providing the training code along with feature extraction code is important to gather interest within the scientific community.

The authors would like to confirm that the code has now been released under AGPL3 license at: <https://github.com/stracquadaniolab/prevent-nf> .

3) Computational details: It's essential to provide specifics regarding the computational procedures. How long does it take to generate FoldX variants for the entire dataset or a subset of 25k data? Additionally, what's the duration of model training? Furthermore, critical aspects such as the choice of hyperparams (such as latent dimension) will be worth reporting. Especially the latent dimension of 128 seems relatively large considering the dataset size and the type of the problem (reconstruction may improve but sampling may become much noisier and harder) -- I suggest ablation study to validate their hyper parameter choices.

We thank the reviewer for the useful comments, and we now added a full hyperparameter analysis in the results section; in particular, we showed that a latent size of 128 is required to obtain a good free energy approximation. Finally, the time required for direct simulations of variants using FoldX is order of magnitudes higher than what required by PREVENT for either training or inference; specifically, the whole training dataset took ~21 days to generate, whereas PREVENT training required ~4h. Despite this is a very favourable comparison, we think it would be misleading to the reader as the hardware used plays a crucial role.

Reviewer #1 (Remarks on code availability):

The authors would like to confirm that the code has now been released under AGPL3 license at: <https://github.com/stracquadaniolab/prevent-nf> .

Reviewer #2 (Remarks to the Author):

****Summary****

In this study, the authors introduce a novel deep learning method, using a Variational Autoencoder (VAE), to approximate the free-energy landscape of proteins. This method, termed PREVENT (PRotein Engineering by Variational frEe eNergy approximaTion), aims to enhance protein thermostability and functionality by generating new, stable, and functional protein sequences. The authors conducted experiments on E. coli phosphotransferase N-acetyl-L-glutamate kinase (EcNAGK) to demonstrate how this free-energy-associated generative model can be beneficial in protein engineering.

****Strengths****

1. The concept of employing VAE to approximate the joint distribution between protein sequences and free energy is well-articulated. VAEs, known for their efficiency in deep learning without requiring an explicit likelihood function (which can be computationally demanding), facilitates the learning of interpretable latent variables, proving to be advantageous for protein engineering tasks.

****Weaknesses****

1. *It is unclear whether the PREVENT-induced mutations are superior to the variants present in the training set. Figure 1.A indicates that the training data's free energy spans from -40 to 150 kcal/mol, while Figure 4.A shows that the generated sequences' free energy peaks at around -30 kcal/mol. If the free energy of generated sequences is not more favorable than that of random mutations, one might question the advantage of PREVENT over extensive random mutation paired with a tool like FOLDX for energy prediction and selection of the highest-energy variants.*

We thank the reviewer for the useful comment, which we addressed in our revised discussion; briefly, simply combining random mutagenesis and FoldX do not scale favourably beyond a handful of mutations, as free energy becomes extremely taxing to compute (~21 days for 100K variants), whereas PREVENT can generate variants in a matter of seconds, including their expected free energy value. Finally, since our models are either directly (sne strategy) or indirectly (poe strategy) seeded around the wildtype protein, it is likely to generate sequences closer, both energetically and in sequence, to the wildtype, rather than an outlier. In practice, this is also a conservative approach, as the free energy estimates for a higher number of mutations are less robust. Finally, it's important to note that generating functional variants in the energetic/sequence neighbourhood of the wildtype is extremely relevant especially in practice, such as in the pharmaceutical industry, where many protein therapies are approved on the basis of non-inferiority, since they allow to broaden the therapeutic offer and usually to lower the costs of the therapy.

2. *The correlation between free energy and functional performance, as tested experimentally, needs clarification. Since the design of sequences heavily relies on predicted free energy, it is crucial to validate if lower energy correlates with enhanced functionality.*

We thank the reviewer for the comment. We have already shown that a) energy driven engineering enables the generation of more diverse proteins and b) the sne2 variant grows faster than the wildtype, so an increase in functionality with respect to the phenotype considered.

3. *The training process details are vague, particularly regarding how the training set was divided and what criteria were used to terminate the training. With 1,400 epochs mentioned, there's a concern that the model might have overfitted the dataset.*

We thank the reviewer for the comment. We have now added a full hyperparameters analysis showing the robustness and efficiency of our model.

4. In the EcNAGK experiment section, the authors mention generating 117,387 unique variants, but the method for generating these variants remains unclear. The method section suggests that variants were randomly generated; it should be specified if this applies to EcNAGK as well and the time required to label these variants with free energy.

We thank the reviewer for pointing this out and we have now clarified the process further in our manuscript. However, we would also like to point out that wall time for generating variants and evaluating their free energy is not a robust metric, since it largely depends on the hardware used.

****Errors****

- Line 306: The symbol ' ϕ ' should be subscripted.
- Line 329: The phrase "seed sequence is passed in input to the decoder" should be corrected to "seed sequence is passed as input to the encoder," as it involves encoding the input sequence to obtain an embedding, not decoding.

All typos have been fixed.

Reviewer #2 (Remarks on code availability):

Did not find code.

The authors would like to confirm that, upon acceptance, we will make the software freely available to the academic at the following url: <https://licensing.edinburgh-innovations.ed.ac.uk>.

Reviewer #3 (Remarks to the Author):

- Can the authors describe better the held-out test set used to evaluate the performance of the generator across the four training set sizes? What size it had, how where the sequence in it selected?

We thank the reviewer for the comments, and we have now further clarified our procedures.

- It would be helpful to present a graphical description of the proposed architecture to help understand how all components fit together.

- "and KL divergence is the Kullback-Leibler." -> "and KL is the Kullback-Leibler divergence."
Fixed.

Finally, I think that there is a missing opportunity in this work which is to dig into the decision making mechanisms of the networks in order to distill (if possible) biological knowledge relevant for the sequence generation and free energy prediction. The use of transformers (with their attention mechanisms) should facilitate this process.

We thank the reviewer for the suggestion but, at this time, we think this is beyond the scope of the paper.

Reviewer #3 (Remarks on code availability):

The authors would like to confirm that the code has now been released under AGPL3 license at: <https://github.com/stracquadaniolab/prevent-nf> .

Response to reviewers

We would like to thank all reviewers for their comments and suggestions. We hereby addressing specific remarks raised by reviewer #3.

Reviewer #3 (Remarks to the Author):

- Can the authors describe better the held-out test set used to evaluate the performance of the generator across the four training set sizes? What size it had, how where the sequence in it selected?

We have added the procedure in both the results section (lines: 80-91) and the methods section (line: 310-320). Please, also note that the model expects a pair sequence-energy, where energy can be obtained by any method the user prefers.

- It would be helpful to present a graphical description of the proposed architecture to help understand how all components fit together.

We have added a schematic of our model in the supplementary material.

- "and KL divergence is the Kullback-Leibler." -> "and KL is the Kullback-Leibler divergence."
Fixed.

We apologized to have missed this typo, but it is now fixed.

Finally, I think that there is a missing opportunity in this work which is to dig into the decision-making mechanisms of the networks in order to distil (if possible) biological knowledge relevant for the sequence generation and free energy prediction. The use of transformers (with their attention mechanisms) should facilitate this process.

We agree with the reviewer about the importance of the question, but we would like to point out that the contribution of our paper does not lie in understanding general design protein engineering rules, but rather on how to approximate sequence and biophysical information with a generative model and then test it experimentally on a non-trivial target; we think the paper achieved this objective. Also determining the best approach to characterise the decision-making process of the model might require further methods development beyond what is available in the explainable AI literature. Nonetheless, we have acknowledged this potential research direction and its importance in our discussion (line: 300-302).